# Smart-Home Concept for Remote Monitoring of Instrumental Activities of Daily Living (IADL) in Older Adults with Cognitive Impairment: A Proof of Concept and Feasibility Study

**DOI:** 10.3390/s22186745

**Published:** 2022-09-07

**Authors:** Myeounggon Lee, Ram Kinker Mishra, Anmol Momin, Nesreen El-Refaei, Amir Behzad Bagheri, Michele K. York, Mark E. Kunik, Marc Derhammer, Borna Fatehi, James Lim, Rylee Cole, Gregory Barchard, Ashkan Vaziri, Bijan Najafi

**Affiliations:** 1Interdisciplinary Consortium on Advanced Motion Performance (iCAMP), Michael E. DeBakey Department of Surgery, Baylor College of Medicine, Houston, TX 77030, USA; 2Neurology and Psychiatry & Behavioral Sciences, Baylor College of Medicine, Houston, TX 77030, USA; 3Menninger Department of Psychiatry and Behavioral Science, Baylor College of Medicine, Houston, TX 77030, USA; 4Michael E. DeBakey Veterans Affairs Medical Center, Houston, TX 77030, USA; 5BioSensics LLC, Newton, MA 02458, USA

**Keywords:** remote patient monitoring, smart home, dementia, aging, Internet of Things (IOT), wearables, digital health, activity of daily living, life-space

## Abstract

Assessment of instrumental activities of daily living (IADL) is essential for the diagnosis and staging of dementia. However, current IADL assessments are subjective and cannot be administered remotely. We proposed a smart-home design, called IADLSys, for remote monitoring of IADL. IADLSys consists of three major components: (1) wireless physical tags (pTAG) attached to objects of interest, (2) a pendant–sensor to monitor physical activities and detect interaction with pTAGs, and (3) an interactive tablet as a gateway to transfer data to a secured cloud. Four studies, including an exploratory clinical study with five older adults with clinically confirmed cognitive impairment, who used IADLSys for 24 h/7 days, were performed to confirm IADLSys feasibility, acceptability, adherence, and validity of detecting IADLs of interest and physical activity. Exploratory tests in two cases with severe and mild cognitive impairment, respectively, revealed that a case with severe cognitive impairment either overestimated or underestimated the frequency of performed IADLs, whereas self-reporting and objective IADL were comparable for the case with mild cognitive impairment. This feasibility and acceptability study may pave the way to implement the smart-home concept to remotely monitor IADL, which in turn may assist in providing personalized support to people with cognitive impairment, while tracking the decline in both physical and cognitive function.

## 1. Introduction

Dementia, a chronic disease of aging, is characterized by progressive cognitive decline that interferes with independent functioning [1,2]. Alzheimer’s disease is the most common type of dementia that refers to a particular onset and course of cognitive and functional decline associated with age [3]. The number of persons with dementia is projected to increase to 75.6 million by 2030 and 135.5 million by 2050 [4], resulting in astronomical economic costs [5]. Effective interventions exist to reduce patient and caregiver strain and psychological morbidity, improve motor and cognitive performance, and delay or prevent admissions to residential care [5,6,7,8,9]. Therefore, early diagnosis of Alzheimer’s disease is pivotal because such interventions are most effective early in the disease course [5,9].

Assessment of activities of daily living (ADL) is essential for timely diagnosing of AD and mild cognitive impairment. Difficulties performing instrumental activities of daily living (IADL) may reflect the symptoms of cognitive decline [10]. The IADL include more complex activities of self-care such as medication management, meal preparation, housekeeping, financial management, shopping, etc. [7,10]. In addition, assessment of global mobility behavior, defined life-space, is also important to identify physical and cognitive functions in older adults [11], and several studies have reported that a limited life-space is related to the potential risk of Alzheimer’s disease, and mild cognitive impairment, as well as the decline of cognitive function in older adults [12,13].

Typically, assessments of IADL and life-space are widely conducted using questionnaires. For instance, Lawton and Brody’s IADL questionnaire is used to assess IADL characteristics, and it is composed of eight items, including telephone use, shopping, food preparation, housekeeping, laundry, mode of transportation, medicine care, and financial management. Each item will be scored if the subject can perform each IADL (yes: 1, no: 0) [14,15]. A shorter life-space questionnaire consists of nine items (yes: 1, no: 0), and each item is asking the subject about a specific area (e.g., “During the past 3 days, have you been to other rooms of your home besides the room where you sleep?”) within the past week [10,16]. Higher values on both questionnaires reflect better functions for IADL and life-space [14,16]. However, patients with severe cognitive impairment may overestimate their IADL capacity because of a lack of awareness of deficits in their abilities [10,17,18,19,20]. Questionnaires are subjective assessments, which may provoke inaccurate results and make it challenging to quantify the level of IADL and life-space [10,17,18].

Recently, several studies have used objective assessment devices to measure the IADL and life-space. For instance, some studies have used monitoring sensors to assess IADL characteristics such as infrared-based [21] and detection sensors with video camera [22]; these technologies may have difficulty distinguishing which subjects are detected among family members, and may cause uncomfortable effects due to recording video in the patient’s environment. In addition, smartwatches and smartphones, including GPS, have been used to measure life-space characteristics [11,23,24]. However, these technologies can only provide the GPS location of the individual; they cannot measure the behaviors of the individual (e.g., physical activity) in the specific location. There is no existing system that can simultaneously measure the IADL and life-space characteristics, and thus, an innovative platform is required to explore objective IADL, as well as life-space characteristics, for diagnosis of dementia severity.

Therefore, the aim of this study is to assess feasibility and validate the novel technology-based platform, called IADLSys, for home-based remote monitoring of IADL and life-space.

## 2. Materials and Methods

### 2.1. Concept and Components: IADLSys

The innovative platform called IADLSys was developed by BioSensics LLC (Newton, MA, USA) to conduct an objective remote-based assessment of IADL and life-space characteristics. This platform can detect the interaction timing and types of IADL in the individual’s environment. IADLSys can also monitor the user’s physical activity and life-space characteristics. This system is composed of three key devices, as follows (Figure 1):
(1)Physical tag (pTAG): A wireless-based sensor that can be attached to various objects to monitor interactions with these objects in the user’s environment(2)Monitoring pendant (PAMSys+): A pendant-type wearable sensor monitors the user’s physical activities, as well as interacts when the user is in proximity to the pTAG.(3)Interactive-tablet PC: The voice-enabled tablet PC can remotely measure and collect questionnaires and other clinical tests such as speech assessment, Trail Making Tests, etc. The collected data can transfer to a secured cloud.

The PAMSys pendant records interactions with the pTAG in the patient’s environment. This information is then relayed to the tablet whenever the user is within range. The tablet will then transfer this data to BioDigit Cloud, a Health Insurance Portability and Accountability (HIPAA)-compliant hosting environment offered by BioSensics LLC for use within clinical trials [25].

### 2.2. pTAGs Locations for Objective IADL Assessment

To collect and analyze the IADL characteristics, the pTAGs attach to objects of interest in the user’s environment. Based on consultations with clinical experts in dementia care (M.E.K. and M.K.Y., co-authors), we placed several pTAGs on different objects of interest to investigate IADLs, including medication management (by attaching the pTAG to a medication box), toileting (by attaching the pTAG on the most used bathroom door), and preparing meals (by attaching the pTAG on the refrigerator door). Other pTAGs were also located based on the user or caregiver’s preferences to explore other functional and pleasurable activities such as going to the backyard, the frequency of using the washing machine, time spent watching TV, the frequency of using the car, etc. (Figure 2). This customization enabled us to minimize the number of needed pTAGs, yet collect the most relevant IADLs representing the daily routine of an individual.

### 2.3. Principle of the Objective IADL Data Acquisition Using IADLSys

Once the pTAGs and monitoring pendant are powered on, then they are ready to record the IADL. The PAMSys+ pendant is the latest wearable sensor that consists of a 3-axis accelerometer, battery, processor, and built-in memory like the earlier version of the pendant sensor (PAMSys™, BioSensics LLC, Newton, MA, USA) [26,27,28,29,30], but it can also record the interactions with the pTAGs. The pTAG is included in the 3-axis accelerometer. When the pTAG moves, the sensor lamp flashes to reflect those accelerations provoked. In addition, if the object with pTAG is activated and provoking accelerations near the pendant, the interaction event will be recorded on the pendant. For instance, if a pTAG is attached to the refrigerator door which is opened or closed by the individual who is wearing the pendant, then this event will be recorded as an interaction event on the PAMSys+ pendant via Bluetooth^®^ (Figure 3).

### 2.4. Data Analysis for the Objective IADL Derived from the IADLSys

The recorded interaction event from the IADLSys can provide raw data via a Microsoft Excel file. The raw data includes several variables such as recorded time (Unix time format), the activated sensor’s ID, movement duration, 3-axis accelerometer (x, y, and z), and received signal strength indication (RSSI). RSSI reflects proximity between pTAG and pendant and is built using several factors from the PAMSys+ pendant (i.e., physical activity patterns) and pTAG signal strength (magnitude and difference). The frequency of the interactions was considered to investigate how many times the individual interacts with the objects of interest over a continuous 7 days. The recorded time value was converted to the real-time format before calculating the frequency of the interaction times. The minimum time interval between prior- and post-frequency of the interaction times was set at 1 h. Based on the recorded data for 7 days, the averaged frequency of the interaction times was calculated.

### 2.5. Validity Tests for Assessing Different Components of IADLSys

This study conducted four sub-tests to verify the validity of the IADLSys (Figure 4).

#### 2.5.1. Simulated Tests to Determine Accuracy of Proximity Detection Algorithm

The aim of test 1 was to investigate proximity detection to evaluate the quality of Bluetooth^®^ connections between the pTAG and the pendant sensor according to the distances. This test was done under 2 conditions: indoor and outdoor (Figure 5). First, the proximity detection test was performed in an indoor environment to confirm the RSSI signal strength between pTAG and PAMSys+. We placed these 2 sensors at a fixed vertical height off the ground (0.762 m), and then placed the sensors with a gradually increased distance between them as follows: 0 m, 0.15 m, 0.30 m, 0.61 m, 0.91 m, 1.22 m, 1.52 m, 1.83 m, 2.44 m, 3.05 m, 4.57 m, 6.10 m, 7.62 m, 9.14 m, 12.19 m, and 15.24 m. We conducted the same proximity detection test in an outdoor, open, flat environment without any large metal or stone surfaces within 1.83 m of the testing area. The distance range was set from 0 to 4 m. RSSI was reported by the Bluetooth^®^ transceiver once every 10 s and the measurement recorded to flash memory for later download and analysis. The radio transmit power level was set at 15 dB. The peak and average RSSI values were calculated.

#### 2.5.2. Testing Accuracy of IADLSys to Detect Simulated IADLs

Test 2 was to conduct 2 sensitivity and specificity tests to demonstrate detection accuracy for the interaction event times by a single subject. For the first test, the 2 pTAGs were attached to explore the accuracy of detecting the type of IADL. Each sensor was attached to an object that exhibited similar motion characteristics to the real IADL: (1) a box mimicking a medicine box; and (2) a cabinet simulating a refrigerator. These sensors were activated 9 and 10 times, respectively (box and cabinet), and the duration of the activation time was 10 to 30 s for each interaction event.

For the second test, the 2 pTAGs were attached to explore the accuracy of detecting according to the movement speeds. One sensor was attached to the cabinet door, which was moved at normal-to-fast speeds, and the other sensor was attached to the room door and was moved at a slow speed.

#### 2.5.3. Acceptability of IADLSys to Monitor Daily Physical Activities: Preliminary Test

Test 3 was to examine feasibility and acceptability of the IADLSys for older adults with cognitive impairment. We recruited 5 ambulatory older adults with clinically confirmed cognitive impairment. We completed questionnaires to assess several functions (1) Lawton–Brody instrumental activities of daily living (IADL) [14]; (2) Montreal Cognitive Assessment (MoCA) to assess cognitive functions [31]; (3) life-space [16]; (4) Center for Epidemiologic Studies Depression (CES-D) Scale to assess the status of the depression [32] and (5) Falls Efficacy Scale-International (FES-I) to assess fear of falling [33]. All participants signed an approved informed consent after reading the study details, and this protocol was approved by the Institutional Review Board (IRB) of Baylor College of Medicine. After educating participants on how to use IADLSys, they were asked to wear PAMSys+ for a duration of one week (24 h/7 days a week). Additionally, they were provided with 5 pTAGs to attach to different objects of interest in their home. Three pTAGS were standardized to attach to their medication box, refrigerator door, and bathroom door. Two pTAGs were personalized to attach to objects the participants claimed to have the most interaction with (e.g., landline phone, TV remote control, backyard door, etc.).

PAMSys+ is using a tri-accelerometer and the same validated algorithms used [26,27,29,30,34,35,36] in the PAMSys™ pendant activity monitoring system (BioSensics LLC, Newton, MA, USA) to remotely monitor physical activities and sleep patterns while wearing this sensor. The sampling frequency for the accelerometer signals was set at 50 Hz [26]. PAMSys+ enables extracting cumulative postures (sitting, standing, lying, and walking) as a percentage out of 24 h, locomotion (daily step count, number and duration of walking unbroken bouts, and cadence), postural transitions (number and duration of sit-to-stand and stand-to-sit postural transitions), activity behavior (sedentary, light, and moderate to vigorous behavior) and sleep (total sleep time, sleep efficiency, etc.).

To determine adherence, we used a prior validated algorithm to monitor adherence for continuously wearing PAMSys [37]. To determine acceptability and perceived ease of use, we used a technology acceptance questionnaire (see Appendix A) and each answer was graded using a Likert scale between 0 (strongly disagree) and 4 (strongly agree). The averages of all scores were reported for determining perceived ease of use; a higher score reflects better acceptability.

#### 2.5.4. Proof of Concept of IADLSys to Detect IADLs of Interest during Activities of Daily Living: Case Report

The last test was to explore the characteristics of IADL using IADLSys for 7 continuous days in 2 older adults with cognitive impairment. We considered 2 representative older adults with cognitive impairment for the case report. The first participant was an older adult with severe cognitive impairment (MoCA < 10), and the second participant’s cognitive functioning was at a mild cognitive impairment level (18 ≤ MoCA ≤ 25) [31]. This study was approved by the IRB of Baylor College of Medicine, and all participants signed their informed consents after reading the study details.

Before set-up for the IADL assessment, all participants and their caregivers were trained on the use of IADLSys and instructed on where the sensors should be located by the research coordinators (A.M., and N.E., co-authors). Several pTAGs were attached to the medicine box, restroom door, refrigerator, dishwasher, landline phone, and back door to detect several IADL characteristics such as medication management, toileting, cooking, housekeeping, and taking a rest in the backyard, respectively. To test the feasibility, we calculated the frequency of the interaction times derived from the IADLSys; and then we calculated averaged and standard error values for the frequency of the interaction times during the 7 continuous days. In addition, we also collected self-report information from participants and their caregivers to compare the differences of the interaction times (e.g., how many times do you go to the restroom per day?/how often do you go to the backyard per week?).

## 3. Results

### 3.1. Results of Test 1: Accuracy of Proximity Detection

Our results confirmed that at distances over 4.57 m, the sensors did not maintain a stable connection via Bluetooth and the RSSI could not be recorded in an indoor environment (see Figure 6a). In addition, our results confirmed that the two sensors did not connect over 3.05 m in outdoor conditions (Figure 6b).

### 3.2. Results of Test 2: Accuracy of IADLSys Detection

Table 1 shows the results of sensitivity and specificity for detection accuracy. Most of the sensors exhibited 100% accuracy for detection of the interaction events attached to the box and cabinet. In addition, the moderate-to-high speed movement condition also exhibited 100% accuracy for detection of the interaction event, whereas the slow speed movement exhibited 83.3% accuracy (sensitivity = 75%, specificity = 100%) for the detection (Table 1).

### 3.3. Results of Test 3: Preliminary Tests of Acceptability on IADLSys

Table 2 summarizes the demographic information of the recruited participants. All participants accepted to wear the PAMSys+ for 7 days continuously, indicating 100% adherence and high feasibility. Figure 7 illustrates the average cumulative postures for all participants (*n* = 5) as durations of sitting (18.19 h, 37.90%), standing (8.69 h, 18.10%), walking (2.26 h, 4.71%), and lying (18.86 h, 39.29%) with a time resolution of 48 h.

Figure 8 shows the spider graph summarizing the acceptability and perceived ease of use results for the IADLSys platform including the use and installation of pTAG at home by participants or their caregivers, as well as wearing PAMSys+ for 24 h/7 days. An average score of 3.17 was obtained for all assessed acceptability scores (agree to strongly agree on acceptability and ease of use), indicating high acceptability. Several sub-item scores exhibited agree to strongly agree levels, including ease of the learning process, easy to work the overall system, and privacy concerns (3 to 4 scores). The other two items exhibited normal agree levels, such as technical support requirements and bothersome wearing the pendant sensor (2.60 and 2.20, respectively).

### 3.4. Results of Test 4: Case Report for IADLSys According to the Cognitive Impairment

The demographic information for the participants with cognitive impairment is shown in Table 3. Participant 1, with severe cognitive impairment, had a moderate fear-of-falling level, less IADL, and limited life-space characteristics (Table 3). This participant exhibited averaged interaction times for 7 days as follows (objective IADL/self-report, *n*/day): (1) medicine box: 2.43/2; (2) restroom: 2.71/2; (3) landline phone: 7.14/4; and (4) back door: 0.71/5 (Figure 9). Participant 2, with mild cognitive impairment, had a high fear-of-falling level and IADL performance characteristics (Table 3). The average interaction times for 7 days were as follows: (1) medicine box: 4.14/4.5; (2) restroom: 4.71/4.5; (3) refrigerator: 3.43/4; and (4) dishwasher 3.86/2.5 (Figure 10).

## 4. Discussion

The aim of this study was to assess feasibility and validate the IADLSys for home-based remote monitoring of IADL and life-space characteristics. We conducted four sub-tests to verify the validity of IADLSys, and we demonstrated the feasibility in an objective home-based remote monitoring system for assessing IADL characteristics. The main findings of this study are as follows: (1) we confirmed the accuracy of proximity detection was successful under 4.57 m indoors and 3.05 m in outdoor environments; (2) the accuracy results of IADL detection were 83.3 to 100% among the objects of interest; (3) participants with cognitive impairment exhibited sedentary lifestyle patterns for 48 h; (4) the preliminary results of acceptability exhibited agree to strongly agree levels for IADLSys; (5) we confirmed the different interaction times patterns according to the severity of cognitive impairment. These findings are discussed in detail below.

### 4.1. Demonstrating Feasibility in IADLSys: Objective Home-Based Remote Monitoring System for Assessing IADL

In this study, to detect IADL of interest an algorithm was designed to detect interaction between the subject, who wore the PAMSys+, and different objects of interest equipped with a pTAG, based on a combination of postural status (walking, standing, sitting, and lying) and detection of the proximity between PAMSys+ and the pTAG of interest. For the communication between PAMSys+ and pTAG, we used Bluetooth, which operates within the industrial, scientific, and medical (ISM) radio band and uses frequency-hopping spread spectrum (FHSS). This technique reduces the likelihood of interference with other appliances such as the microwave or Wi-Fi (2.4 GHz) in households. In addition, pTAGs can broadcast the recorded data across several radio frequency channels for an extended duration in order to mitigate any potential missing interactions. We confirmed the detection algorithm could be accurate when two sensors are getting closer to each other in both indoor (high radio interference) and outdoor (low interference) environments. Therefore, confirmation is required that the subject is interacting with an object (e.g., refrigerator door) tagged with a pTAG sensor.

As we expected, most of the results of sensitivity and specificity of IADL detection exhibited 100% among the objects of interest. However, the sensor attached to a room door with slow speed movement exhibited relatively reduced sensitivity (false negative = three times, 75%) (Table 1), such as when opening the door from the opposite side to the sensor. The room door used in this test was moved slowly when we tried to open and close this door, and this factor may not have been provoking accelerations. In addition, the obstacle between the pTAG and pendant sensor may affect detection of interaction events in the absence of proper acceleration (three times were successfully detected, but the other three times failed). Nevertheless, our results confirmed that detection accuracy under the well-established experimental conditions could provide reliable data to assess IADL characteristics.

The rationale to report the data every 48 h is based on a prior study suggesting that at least 48 h of recording is required to have reliable physical activity data among older adults [36]. Our participants exhibited relatively high sedentary lifestyle patterns for 48 h (total of 77.21% and 37.05 h, combining sitting and lying time), and similar results have been reported by previous studies that the patients with mild cognitive impairment and AD exhibited less physical activity and high sedentary lifestyle patterns [38,39]. These results suggest that assessing physical activity characteristics by remote monitoring systems may be helpful to understand essential life-space characteristics in subjects with cognitive impairment in their living environment. Future studies need to confirm the association between objective physical activity and life-space characteristics.

Additionally, we documented privacy concerns/technology anxiety from the point of view of participants. An average score of 3.17 obtained for all assessed acceptability scores (agree to strongly agree on acceptability and ease of use) indicated high acceptability, except for one subject; the other subject did not voice any privacy concerns or technology anxiety. Among the assessed acceptability sub-scores, the scores of two items were relatively lower than the others, including the perceived need for support from a technical person (score 2.60) and perceived acceptability of wearing PAMSys+ for 24 h/7 days of the week (score 2.20). At the initial baseline visit, research coordinators gave step-by-step instructions and paired the sensors with the system, and with the participants and their caregivers. However, even after the initial support, the participants and their caregivers had some uncertainties, possibly due to the fact this was their first introduction to a smart-home system. The participants were instructed to wear the monitoring pendant for 7 days. Despite the sensor being waterproof and safe to wear during a shower, some participants were not comfortable to wear the sensor during certain activities, including taking a shower/bathing, leading to a lower acceptability score for the pendant sensor being worn 24 h/7 days of week. The other four item scores were higher than 3.00 (range from 3.00 to 4.00 indicating agree or strongly agree). Overall, the results support the feasibility and acceptability of IADLSys to remotely monitor daily physical activity and activities of daily living among older adults with cognitive impairment. The observations should be confirmed with a larger sample size in future studies.

Finally, we conducted a case analysis in two participants with cognitive impairment, and we confirmed interesting results according to the severity of the cognitive impairment. We successfully collected four types of interaction times including for the medicine box, restroom door, landline phone, and backyard door with participant 1 who had a severe cognitive impairment. Interestingly, this participant indicated IADLSys data such as medication management and toileting were relatively in agreement with the self-report. However, this participant underestimated using the landline phone (7.14 times (IADLSys) vs. four times (self-report)), and overestimated spending time in the backyard (0.71 times (IADLSys) vs. five times (self-report)) (Figure 8). Similar results reported that patients with severe cognitive impairment produce biased or inaccurate results for subjective assessment of IADL [10,17,18,19,20,24]. Therefore, the assessment of IADL using remote-based monitoring technology may be required for the objective assessment of IADL, especially in patients with severe cognitive impairment.

We also collected four types of interaction times, including medicine box, restroom door, refrigerator, and dishwasher with participant 2, with mild cognitive impairment. Overall, this participant indicated relatively high agreement between IADLSys data and self-report. In particular, this participant regularly performed the IADLs such as housekeeping (use of dishwasher per day = 3.86 times) and prepared meals (use of refrigerator per day = 3.43 times) (Figure 9). These results suggest that utilizing IADLSys may provide objective and reliable IADL characteristics, which can demonstrate advanced insight into the cognitively impaired patient population in a non-experimental environment.

### 4.2. Strengths and Limitations in This Study

In this study, we proposed the proof of concept of an innovative smart-home platform for identifying interactions between an individual and different objects of interest (e.g., medication box) or locations (e.g., bathroom), which could help to objectively determine IADL including medication management, housekeeping, and leisure activities as well as indoor life-space (the ability to interact with different locations). Using both simulated IADL as well as actual monitoring of IADL in two older adults with cognitive impairment, we demonstrated the accuracy and acceptability of this concept to track activities of interest. Most previous studies pointed out that questionnaire-based assessment provokes inaccurate results [10,17,18,19,20], and it could be difficult to quantify IADL and life-space characteristics in patients with severe cognitive impairment [10,17,18]. IADLSys includes innovative technology to detect and record accurate data collection and is also simple and easy to use. Therefore, we suggest that this innovative platform may explore objective IADL, as well as life-space characteristics, for diagnosis of dementia severity in the individual’s living environment.

However, we also recognized there are several limitations. Firstly, the sample size was limited, early results unveiled an interesting fact that self-reporting of IADL was not accurate among those with significant cognitive impairment, highlighting the limitations of subjective assessment of IADL in older adults. This observation, however, needs to be confirmed in a larger study. Additionally, age-matched healthy participants who have no cognitive impairment are also needed to act as a control group. It may be possible that comparisons between patients with cognitive impairment and healthy control individuals provide significant IADL and life-space characteristics. Therefore, future studies need to consider a large sample size including an age-matched control group.

## 5. Conclusions

The proposed smart-home concept, which is based on using a combination of a pendant sensor and series of smart tags attached to different objects of interest, has several strengths which aid in understanding essential characteristics of the IADL and life-space compared to conventional assessments using questionnaires. In particular, the IADLSys can simultaneously monitor the IADL as well as physical activities characteristics in a non-experimental environment, which can provide advanced analysis in the patients’ living environment. Future studies are warranted to explore the value of the proposed solution to longitudinally track changes over time in IADLs and indoor life-space, which may assist in tracking both physical and cognitive changes over time, as well as identifying the needed support to assist older adults with cognitive impairment to continue to live at home and independently.

## 6. Patents

Part of the algorithm described in this study is protected by a patent pending application owned by Baylor College of Medicine, TX, USA. B.N. is listed as one of the co-inventors (No. 63/009,702).

## Figures and Tables

**Figure 1 sensors-22-06745-f001:**
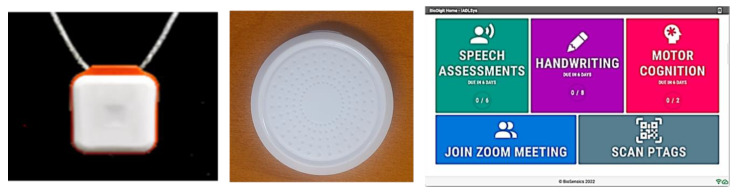
IADLSys: a pendant sensor (PAMSys+, **left**); wireless tag (pTAG, **middle**); and interaction-tablet PC (**right**).

**Figure 2 sensors-22-06745-f002:**
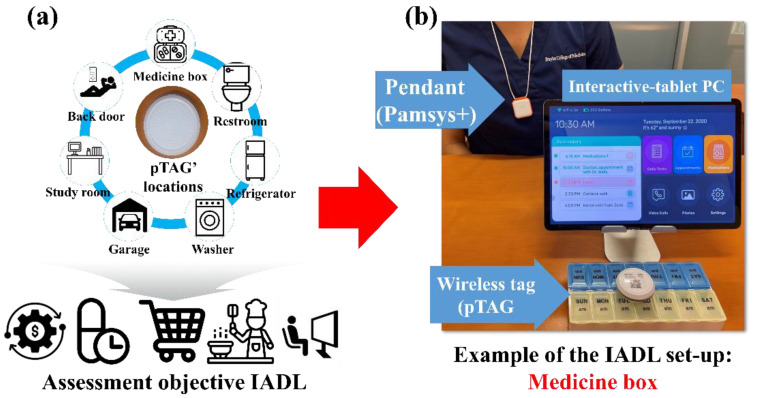
Locations of the pTAGs to assess IADL: (**a**) is describing representative locations for pTAGs; and (**b**) is an example of the objective IADL assessment.

**Figure 3 sensors-22-06745-f003:**
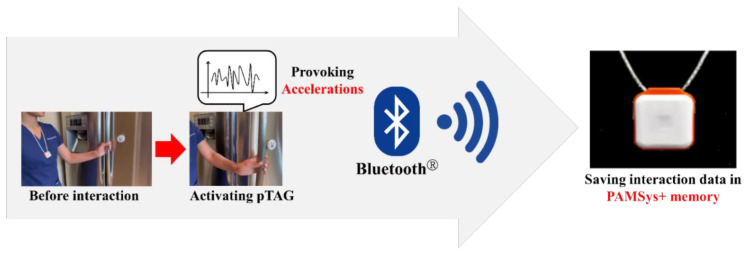
Data acquisition using the IADLSys.

**Figure 4 sensors-22-06745-f004:**
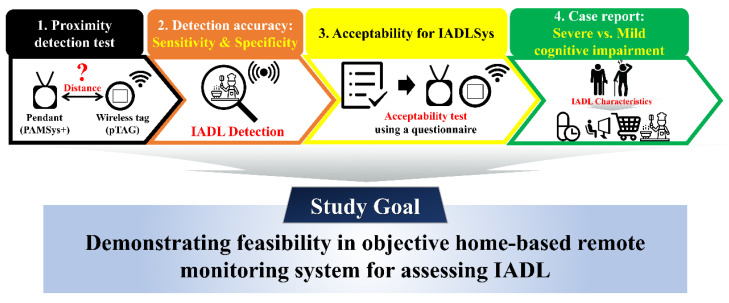
Flow-chart for the validity test.

**Figure 5 sensors-22-06745-f005:**
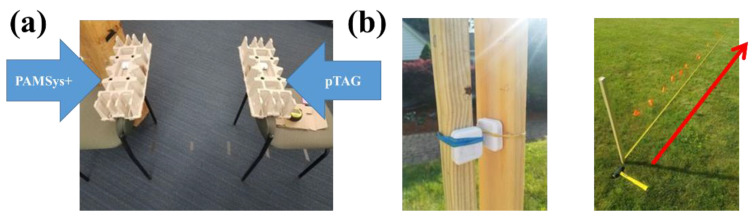
Experimental set-up for proximity detection test: (**a**) was an indoor proximity detection test; and (**b**) was an outdoor proximity test.

**Figure 6 sensors-22-06745-f006:**
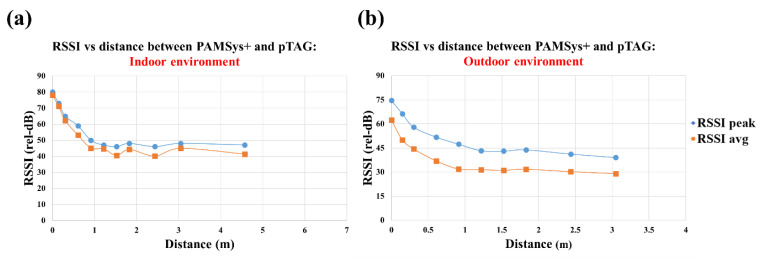
Results of the proximity detection test for the indoor (**a**) and outdoor (**b**) environments.

**Figure 7 sensors-22-06745-f007:**
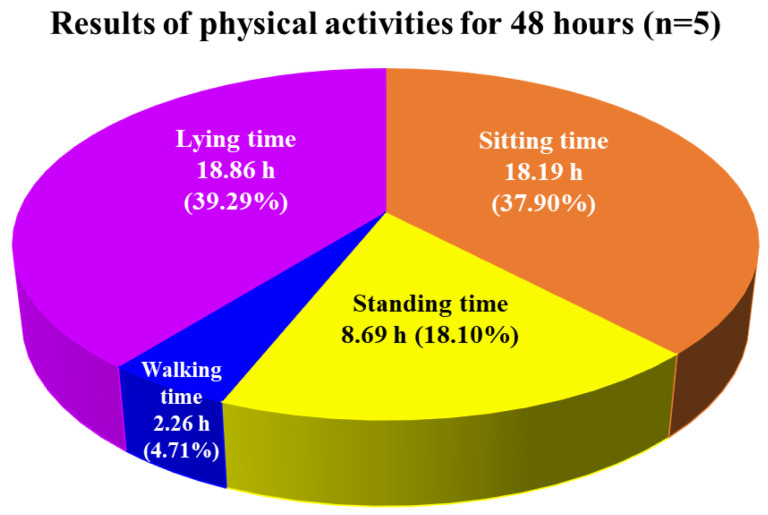
Average of daily physical activities for older adults with clinically confirmed cognitive impairment (*n* = 5), including the average cumulative postures as durations of sitting, standing, walking, and lying time for 48 h by monitoring pendant sensor (PAMSys+).

**Figure 8 sensors-22-06745-f008:**
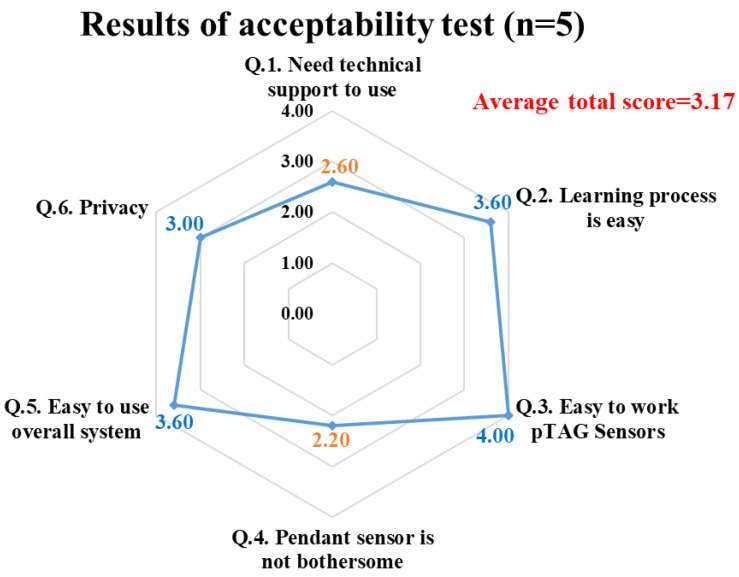
Results of acceptability test for IADLSys.

**Figure 9 sensors-22-06745-f009:**
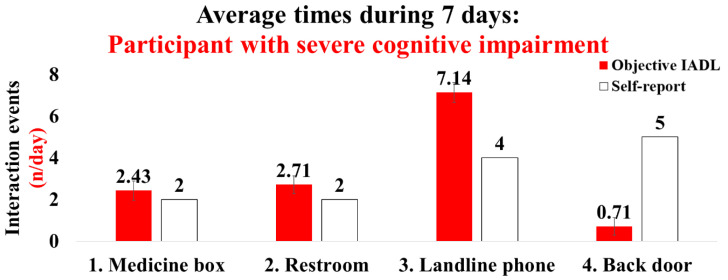
Results of the interaction times for IADLs in the patient with severe cognitive impairment: the objective IADL values provide mean and standard errors for 7 days.

**Figure 10 sensors-22-06745-f010:**
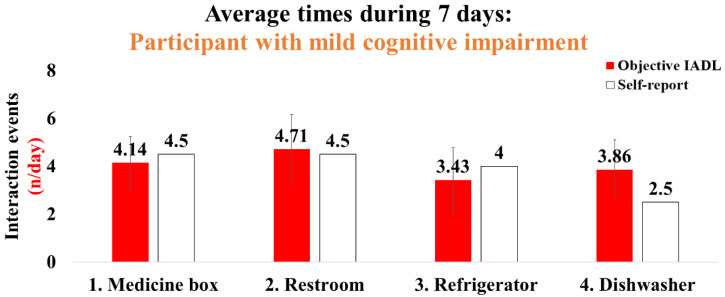
Results of the interaction times for IADLs in the participant with mild cognitive impairment: the objective IADL values provide mean and standard errors for 7 days.

**Table 1 sensors-22-06745-t001:** Results of the sensitivity and specificity for detection of the interaction times according to objects of interest and movement speeds.

	Box (Medicine Box)		Cabinet (Refrigerator)	
	Positive	Negative	Positive	Negative
Positive (Real activations)	True positive *n* = 9	False positive *n* = 0	True positive *n* = 10	False positive *n* = 0
Negative (No activations)	False negative *n* = 0	True negative *n* = 10	False negative *n* = 0	True negative *n* = 9
Sensitivity/Specificity	100%	100%	100%	100%
Accuracy	100%		100%	
	Cabinet door (Moderate to high)		Room door (Slow)	
	Positive	Negative	Positive	Negative
Positive (Real activations)	True positive *n* = 6	False positive *n* = 0	True positive *n* = 9	False positive *n* = 0
Negative (No activations)	False negative *n* = 0	True negative *n* = 12	False negative *n* = 3	True negative *n* = 6
Sensitivity/Specificity	100%	100%	75%	100%
Accuracy	100%		83.3%	

**Table 2 sensors-22-06745-t002:** Demographic characteristics.

Variables	Older Adults with Cognitive Impairment (*n* = 5)
Age, yrs	69.6 (5.4)
BMI, kg/m^2^	26.7 (2.8)
Gender, female %	80%
Cognitive function (MoCA), score	23.0 (1.2)
IADL (Lawton–Brody IADL), score	7.8 (0.4)
Life-space, score	5.2 (0.4)
Depression (CES-D), score	5.0 (5.1)
Concerns for falling (FES-I), score	25.4 (15.1)

Mean (std), BMI: body mass index; CES-D: Center for Epidemiologic Studies Depression Scale; FES-I: Falls Efficacy Scale-International; IADL: instrumental activities of daily living; MoCA: Montreal Cognitive Assessment.

**Table 3 sensors-22-06745-t003:** Demographic characteristics.

Subjects	Age, yrs	BMI, kg/m^2^	MoCA, Score	FES-I, Score	LAWTON, Score	Life-Space, Score
Participant 1 (Female)	82	23.4	6	21	5	5
Participant 2 (Female)	66	20.5	23	33	8	6

BMI: Body mass index; MoCA: Montreal Cognitive Assessment; FES-I: Falls Efficacy Scale-International; LAWTON: Lawton–Brody instrumental activities of daily living scale.

## Data Availability

The datasets are available upon request to the corresponding author.

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
