# Peer review of "Smart-Home Concept for Remote Monitoring of Instrumental Activities of Daily Living (IADL) in Older Adults with Cognitive Impairment: A Proof of Concept and Feasibility Study"

_sensors, 2022, doi:10.3390/s22186745_

Round 1

Reviewer 1 Report

The article is of great interest and well written, I have some suggestions that can be incorporated to this manuscript.

1) Sensors has a Free Format, however all the manuscripts must contain the following items:

  •  Author Information, Abstract, Keywords, Introduction, Materials & Methods, Results, Conclusions, Figures and Tables with Captions, Funding Information, Author Contributions, Conflict of Interest and other Ethics Statements. Check the Journal Instructions for Authors for more details.

 Therefore, I would suggest rearranging the information presented according to the guidelines. 

2) Probably, table 1 and 2 could be integrated in one sole table in order to make the information clearer.

3) In table 3, you are specifying some variables such as Life span score; Depression score and concerns for falling. It would be of interest to know the range of values of each of these variables. 

4) It is important to establish that the results ought to be considered as preliminary because for example in the first experiment only 5 people took place in the experiment; whereas in the second part of the experiment only 2 people participated.

In line 274 you established that the observations should be confirmed with a larger sample, size in the future studies.

Nevertheless, it is important to establish your concerns about the number of participants.

5) In figure 8 and 9, I can observe standard deviation, but I cannot appreciate differences in the size of the standard deviation of the variables studied.

6) I think that the discussions could be extended with a more detailed comparison with the State of Art. 

7) Within the text, there is a contraction and some minor spelling errors should be taken into consideration.

Author Response

The article is of great interest and well written, I have some suggestions that can be incorporated to this manuscript.

Response: We thank your reviewers for their constructive critiques and suggestions. We have highlighted our substantial changes in the revised manuscript. Below we provided responses to each point you raised.

1) Sensors has a Free Format, however all the manuscripts must contain the following items:

  •  Author Information, Abstract, Keywords, Introduction, Materials & Methods, Results, Conclusions, Figures and Tables with Captions, Funding Information, Author Contributions, Conflict of Interest and other Ethics Statements. Check the Journal Instructions for Authors for more details.

 Therefore, I would suggest rearranging the information presented according to the guidelines. 

Response: Thank you for your comments. We entirely revised the manuscript and re-organized the introduction, materials and methods, results, discussion, and conclusion sections. Please confirm the details in the revised version of the manuscript.

2) Probably, table 1 and 2 could be integrated in one sole table in order to make the information clearer.

Response: We have combined the table 1 and 2 based on your recommendation.

3) In table 3, you are specifying some variables such as Life span score; Depression score and concerns for falling. It would be of interest to know the range of values of each of these variables. 

Response: We have added further information and references to explain each assessment and specified important ranges for the assessments. (Please see lines 191-195).

4) It is important to establish that the results ought to be considered as preliminary because for example in the first experiment only 5 people took place in the experiment; whereas in the second part of the experiment only 2 people participated.

In line 274 you established that the observations should be confirmed with a larger sample, size in the future studies. Nevertheless, it is important to establish your concerns about the number of participants.

Response: Thank you for your comment. We have clarified this point and suggested future studies’ direction in the limitations section. 5 older adults were in the experiment, and a case study was completed on 2 of the 5 participants.

5) In figure 8 and 9, I can observe standard deviation, but I cannot appreciate differences in the size of the standard deviation of the variables studied.

Response: The standard error values were only collected by IADLSys because the interaction times reported from the interview were with one value. We have added more clarification in the legend.

6) I think that the discussions could be extended with a more detailed comparison with the State of Art. 

 Response: We have revised the discussion and included essential points.  

7) Within the text, there is a contraction and some minor spelling errors should be taken into consideration.

 Response: A native English speaker has revised the manuscript for grammar, spelling, and punctuation errors.

Reviewer 2 Report

This paper reported the performance evaluation of a remote monitoring system (LADLSys) for the assessment of instrumental activities of daily living of older adults with cognitive impairment. The LADLSys uses three devices to monitor the user’s physical activities and life-space characteristics: a physical tag that can be attached to objects, a monitoring pendant to monitor user’s physical activities, and an interactive-tablet PC for data collection and transmission. The accuracy of proximity detection and simulated IADLs detection was tested. Small clinical studies with 5 older adults with cognitive impairment were also performed. This paper may be published on Sensors after minor revision.

1)     It's not clear how the data communication between the interactive tablet PC and physical tag and monitoring pendant is achieved. What kind of cloud was used fro data storage. Is it HIPAA complied?

2)     The performance of proximity is mainly determined by Bluetooth chips rather than the detection algorithm. 

3)     Bluetooth communication could be interfered by microwave or Wi-Fi in household. These factors should be evaluated in proximity detection test.

4)     It's not clear how the test data from IADLSys can be converted into practical medical outcomes in the home care of older adults with cognitive impairment. Will the data be interpreted by the device or the healthcare professionals? How will the data help with the treatment or management?

Author Response

This paper reported the performance evaluation of a remote monitoring system (LADLSys) for the assessment of instrumental activities of daily living of older adults with cognitive impairment. The LADLSys uses three devices to monitor the user’s physical activities and life-space characteristics: a physical tag that can be attached to objects, a monitoring pendant to monitor user’s physical activities, and an interactive-tablet PC for data collection and transmission. The accuracy of proximity detection and simulated IADLs detection was tested. Small clinical studies with 5 older adults with cognitive impairment were also performed. This paper may be published on Sensors after minor revision.

Response: Thank you for your constructive comments and suggestions. We have carefully considered your comments and suggestions, and we have revised our manuscript accordingly. Below, we provided responses to each point that you raised.

  • It's not clear how the data communication between the interactive tablet PC and physical tag and monitoring pendant is achieved. What kind of cloud was used fro data storage. Is it HIPAA complied?

Response: Thank you for your comments. Yes, the cloud used for recording the IADLSys data is HIPPA compliant. We have added a detailed explanation in the methods section (Please see lines 103-107 in the manuscript).

  • The performance of proximity is mainly determined by Bluetooth chips rather than the detection algorithm. 

Response: Proximity detection is built using several factors within the PAMSys+ including postural status (walking, lying or sitting) and magnitude and change in pTAG signal strength. We have also added this explanation in the methods section (Please see lines 144-146).

  • Bluetooth communication could be interfered by microwave or Wi-Fi in household. These factors should be evaluated in proximity detection test.

Response: We understood what you were concerned about. However, we used Bluetooth operates within the ISM (industrial, scientific and medical) radio frequency band and uses Frequency Hoping Spread Spectrum (FHSS). Therefore, it is tolerant of interference with Wi-Fi and microwave noise. Additionally, most newer Wi-Fi products operate on the 5GHz spectrum which further reduces noise and interference. Finally, physical tags will broadcast data across several radio frequency channels for an extended duration in order to mitigate any potential missed interactions. All testing was done within an open-field environment (ideal) and indoor environment (high radio interference).

4)     It's not clear how the test data from IADLSys can be converted into practical medical outcomes in the home care of older adults with cognitive impairment. Will the data be interpreted by the device or the healthcare professionals? How will the data help with the treatment or management?

Response: We appreciate your comment. We have suggested strengths for the IADLSys platform to assess IADL and life-space characteristics in patients with cognitive impairment in the discussion section.  

Round 2

Reviewer 1 Report

The research is clearly explained, all the suggestions have been addressed.